# Illegal Community Detection in Bitcoin Transaction Networks

**DOI:** 10.3390/e25071069

**Published:** 2023-07-16

**Authors:** Dany Kamuhanda, Mengtian Cui, Claudio J. Tessone

**Affiliations:** 1UZH Blockchain Center, University of Zurich, 8050 Zurich, Switzerland; dany.kamuhanda@uzh.ch; 2Blockchain & Distributed Ledger Technologies Group, Department of Informatics, University of Zurich, 8050 Zurich, Switzerland; 3Department of Mathematics, Science and Physical Education, University of Rwanda-College of Education, Rwamagana P.O. Box 55, Rwanda; 4College of Computer Science and Engineering, Southwest Minzu University, Chengdu 610040, China; mengtian.cui@swun.edu.cn

**Keywords:** bitcoin, transaction networks, blockchain, cryptocurrency, community detection

## Abstract

Community detection is widely used in social networks to uncover groups of related vertices (nodes). In cryptocurrency transaction networks, community detection can help identify users that are most related to known illegal users. However, there are challenges in applying community detection in cryptocurrency transaction networks: (1) the use of pseudonymous addresses that are not directly linked to personal information make it difficult to interpret the detected communities; (2) on Bitcoin, a user usually owns multiple Bitcoin addresses, and nodes in transaction networks do not always represent users. Existing works on cluster analysis on Bitcoin transaction networks focus on addressing the later using different heuristics to cluster addresses that are controlled by the same user. This research focuses on illegal community detection containing one or more illegal Bitcoin addresses. We first investigate the structure of Bitcoin transaction networks and suitable community detection methods, then collect a set of illegal addresses and use them to label the detected communities. The results show that 0.06% of communities from daily transaction networks contain one or more illegal addresses when 2,313,344 illegal addresses are used to label the communities. The results also show that distance-based clustering methods and other methods depending on them, such as network representation learning, are not suitable for Bitcoin transaction networks while community quality optimization and label-propagation-based methods are the most suitable.

## 1. Introduction

Blockchain is a secured, distributed storage of records organized in a chain of blocks of transactions. Records cannot be modified or deleted once they have been added to the blockchain, which ensures data integrity. In permissioned blockchains [1], only authorized individuals can access transactions, whereas in permissionless blockchains such as Bitcoin [2], transactions are publicly available, and anyone can join the network. Bitcoin is a protocol launched in 2009 to facilitate the transfer of its cryptocurrency, Bitcoin (BTC), between users without the need for a central authority, as opposed to other digital payments that are governed by financial institutions [2]. Bitcoin transactions do not reveal personal information about the involved users; instead, users are represented by pseudonymous addresses.

A Bitcoin address consists of at least 26 alphanumeric characters beginning with 1, 3 or bc1 depending on the type of the address: (a) P2PKH, (b) P2SH, or (c) Bech32 [3]. By using such addresses, it is hard to trace the owner. There are concerns that Bitcoin and other cryptocurrency platforms can be used for illegal activities. Researchers and governments are interested in understanding interactions in cryptocurrency transaction networks. Interactions can be modelled as a network (graph) G consisting of a set of vertices (nodes) V and a set of edges E, where $n=\left|V\right|$ and $m=\left|E\right|$. For example, the Bitcoin transaction network can be modelled as a set of transactions: https://www.blockchain.com/explorer/api/blockchain_api (accessed on 28 February 2023) with a directed link from an input transaction to output transactions. It can also be modelled as a network of users where users are vertices, and a link between users indicates the flow of BTC from one user to another. Vallarano et al. [4] used this modelling to investigate the relationship between users’ behaviour and cryptocurrency pricing in exchange markets by focusing on Bitcoin, while De Collibus et al. [5,6] focused on understanding the growth of the Ethereum network. Other works focused on the forensics, such as entity identification [7,8] and using machine learning to detect illegal transactions [9,10,11]. Another way to analyse blockchain transaction networks is to understand the cohesive sub-groups that exist in such networks. Previous work on cluster analysis in blockchain transaction networks focused on address clustering, a task that uses heuristics to group addresses that are assumed to be controlled by the same user [3,12]. Community detection was used to cluster users of similar characteristics [13] but the obtained communities could not be interpreted further as they consisted of pseudonymous addresses. This work combines community detection and off-chain information to quantify the presence of illegal addresses in daily Bitcoin transactions.

This work provides the following contributions: We address the illegal community detection problem in blockchain transaction networks with an emphasis on Bitcoin. Using a set of known illegal addresses linked to illegal darknet markets that have been shut down, we quantify the presence of illegal communities from daily Bitcoin transactions. Illegal community detection is essential in Bitcoin forensics as it helps to identify the addresses most related to known illegal addresses. We benchmark different community detection methods and explain why some are not suitable for Bitcoin transaction networks. From various community detection methods [14,15], we benchmark representative methods from distance-based clustering, network representation learning, spectral clustering, community quality optimization, label propagation and clique-based methods using the conductance, modularity and running time.

The rest of the article is organized as follows: Section 2 discusses the preliminaries, which include blockchain, community, and community evaluation measures. Section 3 discusses materials and methods, which include the data, community detection methods, and the approach used to detect illegal communities. Section 4 consists of experiments and results. Section 5 provides a conclusion.

## 2. Preliminaries

Traditional electronic payments rely on financial institutions as trusted third parties, Bitcoin was launched in 2009 as a decentralized, trustless electronic cash system that offer individuals the ability to transact without a trusted third-party [2]. This was achieved by using a public blockchain to store transactions on a distributed network of nodes and the proof-of-work consensus mechanism to confirm transactions rather than a central authority. When a user initiates a transaction, it is propagated over the network as an unconfirmed transaction. Miners on the network use their powerful computing hardware to compete in solving complex mathematical puzzles (mining) to validate blocks of unconfirmed transactions. Once validated, a block is disseminated over the network and appended to the previous block. The miner of the block receives rewards in the form of minted BTC and transaction fees. Once a block is confirmed and added to the blockchain, its contents become immutable, ensuring the integrity of the recorded data.

While most blockchains store data linearly, emerging ways of storing data using directed acyclic graphs (DAG) have been used [16,17]. A blockchain transaction refers to an action that adds data to the blockchain. It can be a payment in cryptocurrency blockchains; registration or transfer of land ownership in a blockchain-based land management system; or a vote in a blockchain-based voting system, which can be used to ensure the immutability of votes and transparency. As opposed to banking systems or traditional voting systems where users can be identified with personal information, in blockchain-based systems, users use pseudonymous addresses, which make it difficult to trace the owner [7].

User interactions such as friendships in social networks and blockchain transactions are often represented with a network. A network is modelled as a graph G=(V, E) where V is a set of n vertices (nodes): $V=\left\{v_{1}\cdots v_{n}\right\}$ and E a set of *m* edges: $E=\left\{e_{1}\cdots e_{m}\right\}$ ⊆  V × V. Network nodes can be organized into k sets of related nodes known as communities $C=\left\{C_{1}\;\cdots \;C_{k}\right\}$. We define a Bitcoin transaction network as a graph in which nodes are Bitcoin addresses and the edges denote the flow of bitcoins (BTC) between addresses.

Networks usually consist of cohesive subgroups of nodes known as communities. A community is generally defined as a group of nodes that are densely connected between themselves with sparse connections to the rest of the network [18]. This definition focuses on disjointed communities. In overlapping communities, if most of the community members belong to other communities, the overlapping section can become denser than the community. In this case, a community is considered to be a group of nodes with a high probability of being connected to one another than to any other node of the network [14]. An illegal community $C_{i}\in$ C consists of one or more members who have been involved in illegal activities, such as money laundering, scam, terrorism financing and many more.

The conductance of a community $C_{i}$ of a network G, is a community evaluation measure indicating a fraction of community edges that point outside the community [19]:(1)$$\begin{array}{c} \emptyset \left(G,\;C_{i}\right)=\frac{e^{\prime }\left(C_{i}\right)}{min\left(e\left(C_{i}\right),\;\;\;e\left({V-C}_{i}\right)\right)}\; \end{array}$$$e\left(x\right)$ denotes the total number of edges pointing to nodes in x, while $e^{\prime }\left(x\right)$ denotes the total number of edges that link x to the rest of the network. The lower the conductance, the better the community.

The modularity Q measures the strength of the division of a network G into its communities C. As opposed to conductance, the higher the modularity, the better the communities. For disjointed communities, the modularity is defined as [20]:(2)$$\begin{array}{c} Q\left(G,\;C\right)=\frac{1}{2m}\sum_{ij}\left[\left(A_{ij}-\frac{d_{i}d_{j}}{2m}\right)\delta \left(h_{i},h_{j}\right)\right] \end{array}$$
where A is the adjacency matrix, $d_{i}$ is the degree of the node i, m is the number of edges of the network and $h\in N^{n}$ is a community assignment vector for n nodes. The function $\delta \left(h_{i},h_{j}\right)$ is 1 if the community assignment value $h_{i}$ for node i is the same as the community assignment $h_{j}$ for node j. An adapted version has been proposed for overlapping communities [21]:(3)$$\begin{array}{c} Q\left(G,\;C\right)=\frac{1}{k}\sum_{i=1}^{k}q\left(C_{i}\right)\; \end{array}$$
(4)$$\begin{array}{c} q\left(c\right)=\frac{1}{n_{c}}\sum_{i\in c}\frac{\sum_{j\in c,\;j\neq i}A_{ij}-\sum_{j\notin c,}A_{ij}}{d_{i}\;\cdot \;k_{i}}\;\cdot \;\frac{m_{c}}{\left(\begin{array}{c} n_{c}\\ 2 \end{array}\right)}\;\; \end{array}$$
where k is the number of communities in G; $k_{i}$ the number of communities containing node i; $d_{i}$ the degree of a node i; $n_{c}$ the number of nodes in the community c; $m_{c}$ the number of edges in the community c and $A_{ij}$ an entry in the adjacency matrix A for nodes i and j.

## 3. Materials and Methods

Bitcoin transactions are publicly accessible using blockchain data APIs: https://www.blockchain.com/explorer/api/blockchain_api (28 February 2023). As a single user often owns multiple addresses, heuristics are used to create a user transaction network (user detection) from the original transaction network. Community detection can be used to group related network users into clusters, which can be labeled using off-chain information to interpret what the communities represent.

### 3.1. Illegal Community Detection

We use community detection to detect a group of users associated with illegal Bitcoin addresses. In the cryptocurrency sense, we define a community as a group of users who often send coins between themselves, rather than conducting transactions with the rest of the network. Ground-truth illegal communities could be useful when evaluating the performance of algorithms in detecting illegal communities, but such labeled communities are not available for blockchain transaction networks. However, the known illegal addresses could be used to identify illegal communities, which help to identify more users that are likely to be associated with illegal activities.

We used a two-step approach: (1) we used a community detection method to detect unlabeled communities; (2) then, we used known illegal addresses to label the detected communities. In the first step, we initially benchmarked the representative algorithms discussed in Section 3.4 on Bitcoin user transaction networks in Table 1. The networks were obtained by applying the multi-input heuristic on raw Bitcoin transactions. For each network in Table 2, we used the selected algorithm with the best modularity, average conductance, and speed results to detect the unlabeled communities. In the second step, we iterated through the detected communities, checking the presence of a user with illegal addresses linked to darknet markets in Table 3. Some communities consist of users associated with many illegal addresses, while other communities are associated with one illegal address. Although illegal activities such as money laundering can be carried out by one community member (by sending BTC to addresses he/she controls), other members that often transact with the member might be subject to investigation or become victims. For this reason, we mark the entire community as illegal.

### 3.2. Data

We used the Blockchain Data API (https://www.blockchain.com/explorer/api/blockchain_api) to obtain blocks of transactions on different dates, then parse the blocks to construct transaction networks. The multi-input heuristic (Section 3.3) is used to construct user transaction networks, as shown in Table 1 and Table 2. To detect illegal communities, we use networks constructed on dates where illegal activities were detected and reported. Based on the money laundering reported by the Federal Bureau of Investigation (FBI) and Homeland Security Investigations (HSI), as discussed in the Case 1:22-mj-00022-RMM [22], we selected blocks of transactions for the 25 January 2017, 26 January 2017 and 27 January 2017, in which money laundering took place. We also randomly selected a recent date 17 January 2023, then construct networks on the selected dates using multi-input heuristics.

Community detection alone does not provide meaningful information, as community nodes are just pseudonymous addresses. Using the Wallet Explorer API: https://www.walletexplorer.com/ (accessed on 21 October 2022), the IKNAIO API (https://www.ikna.io/) and darknet markets, we selected 13 illegal darknet markets that were shut down and collect the associated 2,313,344 Bitcoin addresses, which we consider to be illegal addresses. Wallet Explorer maps entities to Bitcoin addresses used before 2018 [8,23], while IKNAIO is a CryptoAsset Analytics tool with an API that can be used to access its data. Both data sources have been used for data validation.

### 3.3. User Detection

Multi-input, peel chain and change address identification heuristics were previously proposed for user detection tasks [3,24]. A peel chain is a transaction pattern, which reduces the large amount of cryptocurrency associated with an address through a series of small transactions [3]. The change is the amount that goes back to the sender, who initiated the transaction, and different heuristics have been proposed to detect this [3,24,25]. The multi-input heuristic assumes that multiple input addresses to a transaction are controlled by the same user, as the user collects coins from different addresses to make a payment of a greater amount than that in each of the input addresses [3,24]. Multi-input heuristics are the most consistently used in the literature. We use this in this work to create user transaction networks.

### 3.4. Community Detection

Different community detection methods exist [14,26]. We investigate and group different methods into six categories based on the techniques used to detect communities. From each category, we select popular methods based on their performance on other social networks. We benchmark them on Bitcoin transaction networks to select a suitable method for illegal community detection.

#### 3.4.1. Distance-Based Methods

K-means [27], DBSCAN [28] and hierarchical clustering [29] are popular distance-based clustering methods. Specifically, k-means and DBSCAN are the most used for cluster analysis in Bitcoin transaction networks [13,24]. We select k-means and DBSCAN from this category for our benchmark.

#### 3.4.2. Spectral Clustering

Spectral clustering starts by computing the Laplacian matrix L=D−A or its variants [30], where D is a degree matrix. The next step computes the eigenvectors of L, which can be achieved using eigenvalue decomposition, to find an approximation of L as $L≅\;U\Lambda U^{T}$. The columns of U are eigenvectors of L, while Λ consists of eigenvalues on the diagonal. The third step extracts from U a matrix H, corresponding to the top k eigenvalues, where each row represents a vertex’s representation in k dimensional space. Then, a clustering algorithm such as k-means is applied to H to group its vertices into k communities. The number of communities k is often estimated using eigengap heuristic, where k is related to the largest gap between consecutive eigenvalues $\lambda_{k\;}$ and $\lambda_{k+1}$ of L compared to gaps between $\lambda_{1}$, $\lambda_{2}$… $\lambda_{k\;}$ [31].

#### 3.4.3. Community Quality Optimization

This category consists of methods that find communities that maximize the modularity [20], minimize the conductance [32], or optimize any other community quality measure. Modularity maximization is the most popular. This method aims to split the network into communities that maximize the modularity measure in Equation (2). Leiden [33] and Louvain [34] are recent popular algorithms based on the modularity maximization, while the first modularity maximization method was introduced by Newman and Girvan [20]. Louvain combines the label propagation method (Section 3.4.4) with the modularity optimization for a fast community detection that is scalable to large networks. We selected Leiden and Louvain for global community detection, and Personalized PageRank (PPR) and heat kernel (HK) [32] to optimize the conductance in Equation (1) for local community detection.

#### 3.4.4. Label Propagation

Label propagation (LP) assumes that a vertex v chooses to be in a community to which most of its neighbors belong [35]. LP starts by ranking all vertices of the input network in a certain order, and each vertex is assigned a unique label. From various iterations, labels propagate through the network until each vertex has a community label, carried by the maximum number of neighbors. As all labels are unique at the first iteration, the first vertex takes a random label from its neighbors and, in the subsequent iterations, ties are broken uniformly and randomly. The adapted label propagation methods have been proposed to handle community detection in dynamic networks [36], to improve the speed [37] and detect overlapping communities [38]. The LP [39] was selected for disjoint community detection, while DEMON [40] was selected for overlapping community detection.

#### 3.4.5. Clique Percolation Method (CPM)

CPM finds communities based on adjacent cliques of size k [41]. A clique is a set of vertices that are completely connected to one another. Two cliques of size k are adjacent if they share k−1 vertices. As the detected communities depend on the clique size k being used as input, these communities are known as k-communities. For example, one may find communities based on adjacent 3-cliques or 5-cliques. We used a 2-cliques approach in our benchmark to capture communities consisting of two users who often transact between themselves.

#### 3.4.6. Network Representation Learning

This category consists of matrix-decomposition-based methods: PCA, SVD [42], NMF [43]; graph-diffusion-based methods: LINE [44], DeepWalk [45], Node2Vec [46]; Autoencoder [47], and graph neural networks [48]. These methods aim to learn a representation of the network that captures similarities between nodes better than the adjacency matrix. Apart from NMF, which can output communities without requiring an additional clustering method [49], other methods in this category require that the obtained embeddings are clustered using distance-based clustering methods. Some methods, such as Cluster-GCN include a clustering layer, but require that the number of clusters is specified [50]. We selected Node2Vec due to its popularity for generating embeddings, and DBSCAN to cluster the embeddings into disjoint communities. We also selected BigClam [51], a popular matrix decomposition-based method for overlapping community detection.

## 4. Experiments and Results

From the different community detection methods that exist, this section aims to investigate the most suitable methods for Bitcoin transaction networks. We also investigate the presence and interactions of 2,313,344 known Bitcoin illegal addresses in daily Bitcoin transactions.

We compared methods that detect communities of the same category using community quality measures and execution time: (1) Leiden, Louvain, label propagation, Node2Vec followed by DBSCAN, and spectral clustering with *k*-means are compared in terms of their ability to detect disjoint communities in the entire network; (2) clique percolation method (CPM), DEMON and BigClam are compared for their ability to detect overlapping communities; (3) heat kernel (HK) and personalized PageRank (PPR) for their ability to detect local communities to which a seed node of interest belongs. For local community detection, we randomly selected 100 seed nodes and averaged the obtained scores. The modularity measure is only computed on global communities, as it is a global measure, and the average conductance is used to measure the quality of both global and local communities. In the benchmarked algorithms, Node2Vec with DBSCAN denotes that DBSCAN was used to cluster the embeddings generated by Node2Vec. Spectral clustering with k-means denotes that k-means was used to cluster the eigenvectors obtained from the Laplacian matrix. Eigengap heuristic was used to estimate k. Default parameters were used, except for CPM, where 2-cliques were selected to capture the communities consisting of two isolated users who only transacted between themselves.

### 4.1. Results

The results show that the structure of Bitcoin transaction networks consist of disconnected components as shown in Figure 1.

#### 4.1.1. Benchmark Results

The benchmark of different community detection methods shows that distance-based methods struggle to detect communities on Bitcoin transaction networks that consist of disconnected components. Network representation learning methods also suffer from their dependence on distance-based methods in the clustering phase.

Algorithms that return communities with the highest modularity (Figure 2) also have the lowest conductance (Figure 3 and Figure 4), which indicates that the detected communities are dense, with few or no connections to the rest of the network. This can occur when a dense component is detected as a community. Leiden, Louvain, and label propagation outperformed spectral clustering with k-means and Node2Vec with DBSCAN in detecting densely connected disjoint communities based on modularity and average conductance scores. For global overlapping community detection, the clique percolation method (CPM) outperformed other algorithms from $G_{1}$ to $G_{7}$ but did not scale beyond networks with more than 300,000 nodes and its results from $G_{8}$ to $G_{11}$ were empty (Figure 2).

By analyzing the speed of the algorithms (Figure 5), algorithms that learn the embedding of the network or estimate the number of communities are the slowest. Leiden, Louvain, and label propagation (LP) are the fastest methods and do not use any of those steps. Given their good performance results in detecting dense communities, they are the most suitable for Bitcoin transaction networks.

#### 4.1.2. Illegal Community Detection Results

Based on the performance of Louvain in the first experiment, we used it to detect communities that were labeled using known illegal addresses (Table 3) to detect illegal communities. The results show that 0.06% of the detected communities are illegal, with one or more illegal addresses. Figure 6 shows a community from the transactions of 25 January 2017 ($G_{12}$) containing 19 users associated with 69 illegal addresses.

Figure 7 shows ranked communities by size, with associated conductance (left) and size (right). Red markers represented illegal communities. We identified some illegal communities that were isolated from the rest of the network, as indicated by their zero conductance.

### 4.2. Discussion

As community detection with spectral clustering and Node2Vec consists of two phases, embedding and clustering, we investigated the phase that contributed to the poor community detection results. We identified that the embedding phase could learn representations with visible clusters (Figure 8) and the clustering phase was the issue (Figure 9). Figure 8 shows the embedded representation of $G_{2}$ presented in Figure 1 (right section) with visible clusters while Figure 9 shows poor performance of k-means with different values of k.

The same problem was identified with DBSCAN. By using a grid search that aimed to find the best combinations of its parameters, we could not find any combination that could generate good clusters on bitcoin transaction networks, as shown in Figure 10.

## 5. Conclusions

Community detection can be used in blockchain transaction networks to detect illegal communities containing users associated with illegal activities. It becomes easy to identify other users in the community that are likely to be involved in illegal activities based on their interactions in the community. However, the lack of illegal ground-truth communities makes the illegal community detection task challenging in terms of validating the detected communities. We addressed the challenge by using known illegal addresses to label the detected communities. By using 2,313,344 Bitcoin addresses associated with illegal darknet markets and a community detection method selected based on the results of the benchmark, 0.06% of the detected communities on daily transaction networks contain one or more illegal addresses involved in illegal transactions. Illegal transactions include the purchase of illegal products and money laundering.

The results also showed that distance-based clustering methods, specifically k-means and DBSCAN, did not yield good results in our benchmark of different community detection methods on Bitcoin transaction networks. Spectral clustering and network representation learning-based methods are affected depending on the k-means or DBSCAN in their clustering phase. Adapting distance-based clustering methods to improve their performance on disconnected networks would improve their performance on blockchain transaction networks. Obtaining a larger dataset of illegal addresses could help to identify more illegal communities from daily transactions.

## Figures and Tables

**Figure 1 entropy-25-01069-f001:**
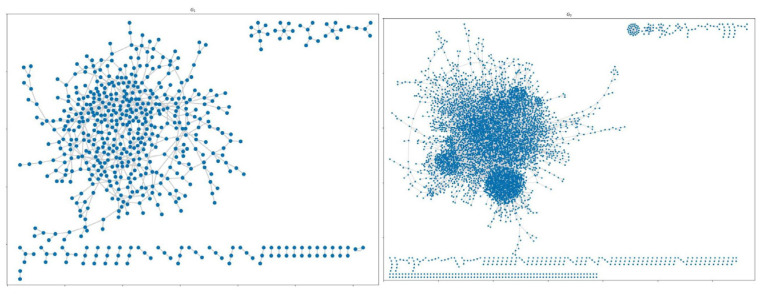
Structure of Bitcoin transaction networks ($G_{1}$ and $G_{2}$). The networks consist of disconnected components and this pattern remains the same in all analyzed Bitcoin transaction networks.

**Figure 2 entropy-25-01069-f002:**
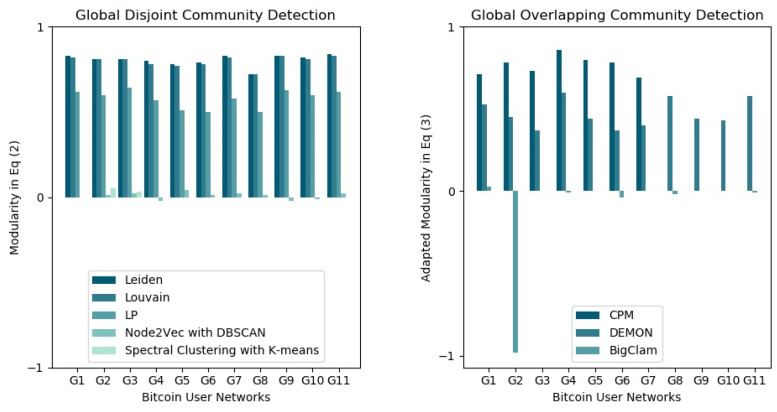
Modularity results. Modularity in Equation (2) was used to evaluate the quality of global disjoint communities, while the modularity in Equation (3) was used for global overlapping communities.

**Figure 3 entropy-25-01069-f003:**
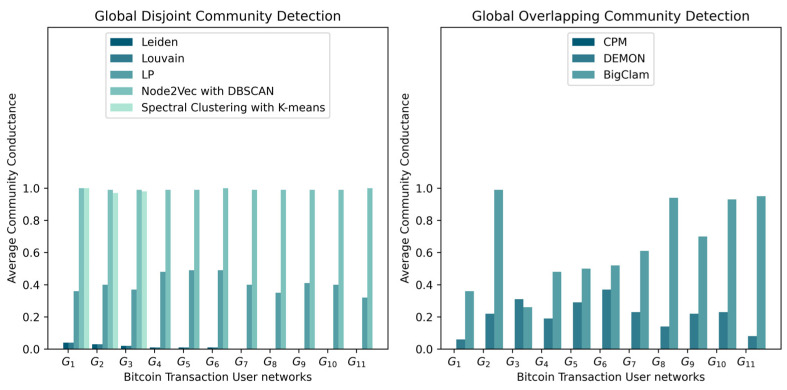
As the conductance measures a fraction of community edges that points outside the community, the conductance consisting of zeros for CPM indicates that all the detected communities have very few or no edges pointing outside the community.

**Figure 4 entropy-25-01069-f004:**
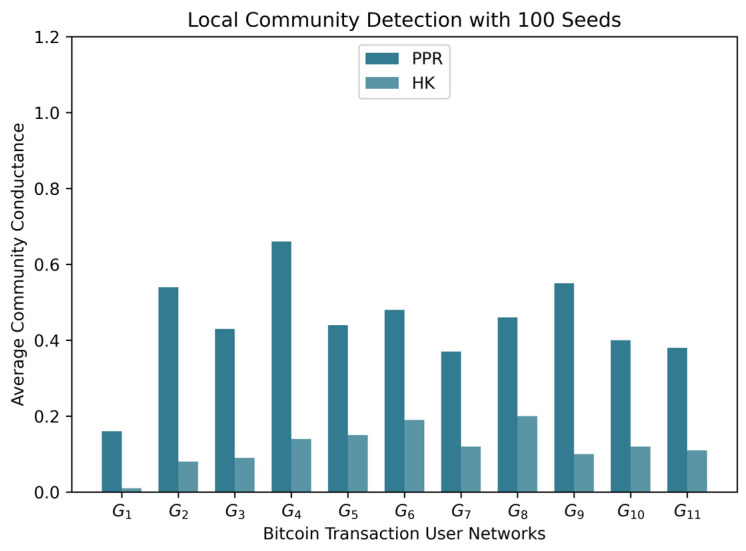
Average conductance results computed on the communities of 100 Bitcoin users. Heat kernel (HK) outperformed personalized PageRank (PPR) in detecting communities that do not have many edges pointing outside.

**Figure 5 entropy-25-01069-f005:**
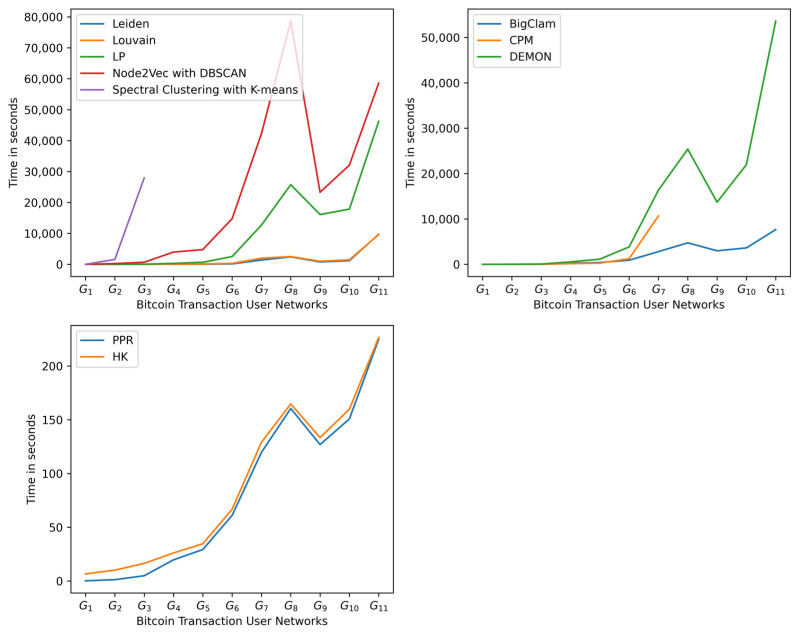
Most of the algorithms that perform well in the detection of good community structures are the fastest, except for heat kernel (HK).

**Figure 6 entropy-25-01069-f006:**
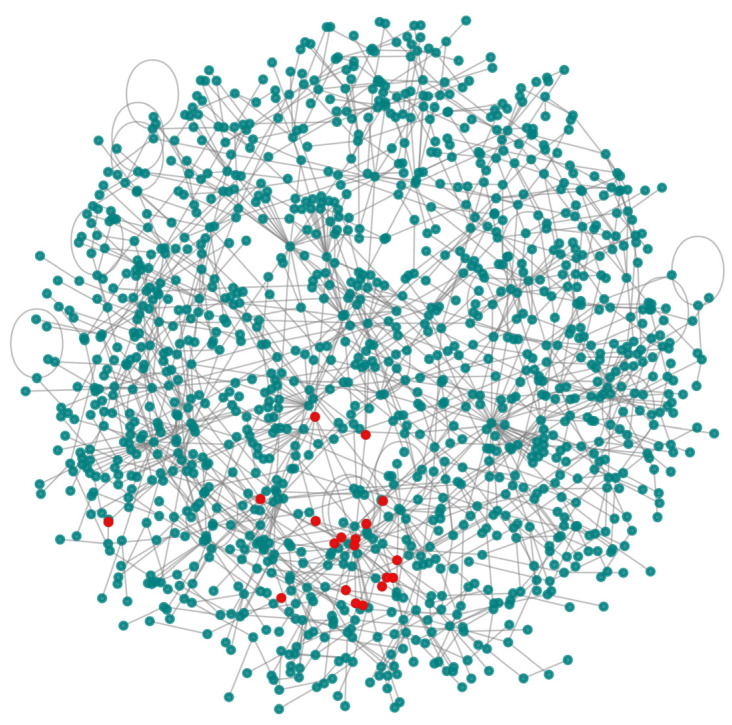
An illegal community consisting of 19 illegal users. Most of these users interact between themselves, as shown by red nodes.

**Figure 7 entropy-25-01069-f007:**
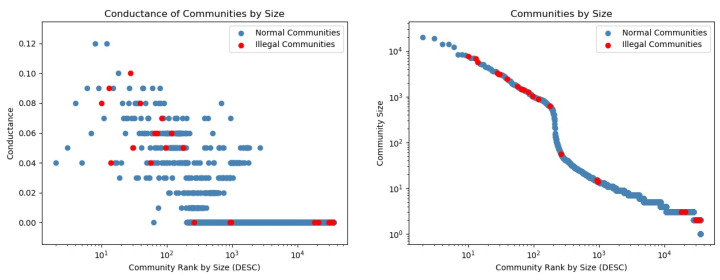
Conductance and size of the detected communities in $G_{12}$. Red markers indicate illegal communities. We identify large communities with a low conductance (<0.5), which is not usually the case in other social networks. Regarding the distribution of community sizes (right), the steep drop after $10^{2}$ on the x-axis and close to $10^{3}$ on the y-axis indicates that few communities (around 100) are very big, and each of the remaining communities consist of less than 1000 members (the network consisted of 35,296 communities in total).

**Figure 8 entropy-25-01069-f008:**
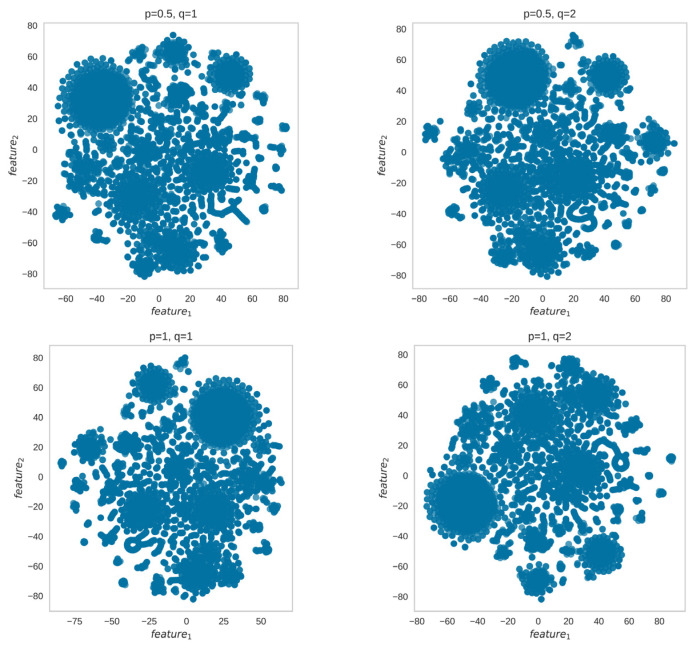
The embeddings learned by Node2Vec on the network shown in Figure 1 contain visible clusters for different parameters. However, the clustering phase with k-means or DBSCAN could not detect clusters with good modularity or average conductance, as shown by the low modularity and high conductance results in Figure 2 and Figure 3 on $G_{2}$.

**Figure 9 entropy-25-01069-f009:**
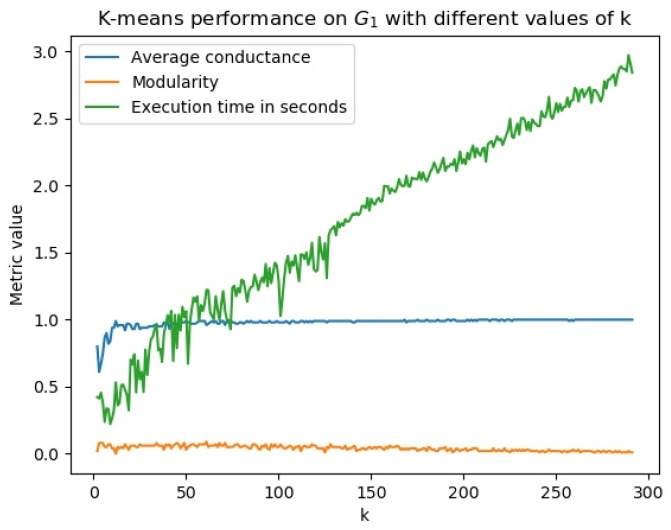
Iterating from k=2 to k=292 (half of the network’s total number of nodes of $G_{1}$), clusters detected have a modularity close to zero and an average conductance close to 1, which indicates a bad community structure.

**Figure 10 entropy-25-01069-f010:**
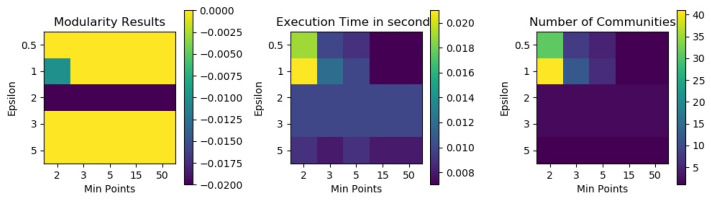
DBSCAN results in terms of modularity, number of detected communities and execution time for different combinations of epsilon and min points on $G_{1}$. All modularity values are very low (zero or less), which indicates a bad community structure.

**Table 1 entropy-25-01069-t001:** Bitcoin transaction networks we used to investigate the structure of Bitcoin transaction networks and suitable community detection methods.

Bitcoin User Transaction Networks	Nodes	Edges
$G_{1}$: 31 December 2010	584	667
$G_{2}$: 31 December 2011	5976	7553
$G_{3}$: 31 December 2012	17,594	29,730
$G_{4}$: 31 December 2013	56,454	93,612
$G_{5}$: 31 December 2014	81,490	127,796
$G_{6}$: 31 December 2015	147,813	238,988
$G_{7}$: 31 December 2016	285,391	421,900
$G_{8}$: 31 December 2017	365,761	557,759
$G_{9}$: 31 December 2018	300,135	418,657
$G_{10}$: 31 December 2019	346,181	507,036
$G_{11}$: 31 December 2020	516,398	666,037

**Table 2 entropy-25-01069-t002:** Bitcoin user transaction networks we used to detect illegal communities.

Bitcoin User Transaction Networks	Nodes	Edges
$G_{12}$: 25 January 2017	711,255	1,376,711
$G_{13}$: 26 January 2017	720,994	1,560,367
$G_{14}$: 27 January 2017	645,053	1,400,766

**Table 3 entropy-25-01069-t003:** Darknet markets and their bitcoin address count.

Darknet Market	Bitcoin Address Count
AlphaBayMarket	189,776
SilkRoad2Market	350,036
SilkRoadMarketplace	372,753
YABTCL.com	3243
AgoraMarket	498,001
SheepMarketplace	53,639
BlackBankMarket	50,878
PandoraOpenMarket	55,757
NucleusMarket	146,381
BlueSkyMarketplace	18,997
MiddleEarthMarketplace	34,149
AbraxasMarket	119,119
EvolutionMarket	420,615
Total	2,313,344

## Data Availability

Bitcoin transaction data is publicly available. We used the Blockchain Data API (https://www.blockchain.com/explorer/api/blockchain_api) to get blocks of transactions for a particular date, then parsed the blocks to construct transaction networks. Bitcoin addresses associated with darknet markets were obtained using the Wallet Explorer API (https://www.walletexplorer.com/) and validated using data from IKNAIO.

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
