# Peer review of "Illegal Community Detection in Bitcoin Transaction Networks"

_entropy, 2023, doi:10.3390/e25071069_

Round 1

Reviewer 1 Report

This paper studies the problem of illegal community detection in bitcoin transaction networks. It is claimed that the illegal community detection task meet the challenge of validating the detected communities. Therefore, the authors propose a method which benchmarks existing community detection methods based on community quality measures to select the best method for detecting unlabeled communities. They first use a community detection method to detect unlabeled communities, and then use known illegal addresses to label the detected communities.

Strengths:

1. They found that adapting distance-based clustering methods to disconnected networks would improve their performance on blockchain transaction networks.

2. The experiments conducted over dataset show positive results.

3. The overall presentation and organization are good.

Weaknesses:

1. The experimental part need be enhanced. The authors analyze the properties of different community detection methods. The number of experimental method comparisons is small, and the credibility of the experiment needs to be increased.

2. The calculation cost and universality of the method proposed in this paper should be considered.

3. The discussion of related work is a little short, the authors should add more in it, especially leveraging large deviation theory or machine learning to detect illegal transactions and community, e.g.,

[1] Securing IOTA blockchain against tangle vulnerability by using large deviation theory, IEEE IoT-J, 2023.

[2] Toward prevention of parasite chain attack in iota blockchain networks by using evolutionary game model, Math, 2022.

[3] Detection of illegal transactions of cryptocurrency based on mutual information, electronics, 2023.

[4] Illegal activity detection on bitcoin transaction using deep learning, Soft Computing, 2023.

Author Response

Reply to Reviewer #1

We would like to thank you for recognizing the strength of the paper and identifying the weaknesses. We have taken the comments into consideration when revising the manuscript:

Comment 1: Strength:

  1. They found that adapting distance-based clustering methods to disconnected networks would improve their performance on blockchain transaction networks.
  2. The experiments conducted over dataset show positive results.
  3. The overall presentation and organization are good

Reply:  Thank you for the positive comments that really encourage us.

Comment 2: The experimental part need be enhanced. The authors analyze the properties of different community detection methods. The number of experimental method comparisons is small, and the credibility of the experiment needs to be increased.

Reply: Thank you for the comment. We have enhanced the experiments section by clarifying how we benchmarked 10 community detection methods that solve 3 different problem formulations of community detection:  (1) Leiden, Louvain, Label Propagation, Node2Vec followed by DBSCAN, and Spectral Clustering with k-means are compared for detecting disjoint communities of the entire network; (2) Clique Percolation Method (CPM), DEMON and BigClam are compared for detecting overlapping communities; (3) Heat Kernel (HK) and Personalized PageRank (PPR) for detecting local communities to which a seed node of interest belongs.

Comment 3:  The calculation cost and universality of the method proposed in this paper should be considered.

Reply: Thank you for the comment. The community detection methods benchmarked are popular in the literature. The approach we used to label illegal communities is our own idea. The darknet markets used have been shut down which can be verified. For example, Alphabay market was shut down in 2017: https://www.justice.gov/opa/pr/alphabay-largest-online-dark-market-shut-down.

SilkRoad 2, Pandora and other markets: https://www.justice.gov/usao-sdny/pr/dozens-online-dark-markets-seized-pursuant-forfeiture-complaint-filed-manhattan-federal

The addresses of these markets has been gathered from 2 different sources WalletExplorer API and IKNAIO for comparison and validation.

Comment 4: The discussion of related work is a little short, the authors should add more in it, especially leveraging large deviation theory or machine learning to detect illegal transactions and community, e.g.,

[1] Securing IOTA blockchain against tangle vulnerability by using large deviation theory, IEEE IoT-J, 2023.

[2] Toward prevention of parasite chain attack in iota blockchain networks by using evolutionary game model, Math, 2022.

[3] Detection of illegal transactions of cryptocurrency based on mutual information, electronics, 2023.

[4] Illegal activity detection on bitcoin transaction using deep learning, Soft Computing, 2023.

Reply: Thank you for suggesting additional references. They have been added with a discussion of using machine learning for illegal transaction detection and IOTA as an emerging ledger that stores data using Directed Acyclic Graph as opposed to Bitcoin which uses a linear blockchain.

Reviewer 2 Report

Illegal Community Detection in Bitcoin Transaction Networks 

The paper focuses on detecting illegal communities in Bitcoin transaction networks. The research investigates the structure of Bitcoin transaction networks and appropriate community detection methods. 

The Abstract needs improvement. It should emphasise more on the challenges so that the readers can get an overview of why this research work is necessary. 

The Introduction is satisfactory. However, the first paragraph requires more clarification.  

“Although the technology can be used in various domains, it is most used in cryptocurrency”

--> To clarify, the blockchain has been used in many domains for a few years now. 

“Blockchain transactions are publicly available, but users are linked to pseudonymous addresses which make them often remain unknown.” 

--> A transaction does not always mean a cryptocurrency transaction. In a permissioned blockchain, the transactions are not publicly available. Thus, it would be nice to indicate the difference between different types of blockchains. 

Section 2 needs improvement with more context. 

A background section is needed to define the concepts more clearly. 

Section 4 requires more clarification on the experiments and results. Also, the Figures are too small and need improvement. 

Such a broad definition of the Conclusion is not necessary. The Conclusion should be concise and solely focus on discussing the research output and future scope of the work. 

Overall, the paper lacks clarity. The authors need to define the concepts and methods more clearly for readers to understand the contribution fully. 

 it is most used in → it is mostly used in

with emphasis on Bitcoin → with an emphasis on Bitcoin

Author Response

Reply to Reviewer #2

Comment 1:  The Abstract needs improvement. It should emphasise more on the challenges so that the readers can get an overview of why this research work is necessary.

Reply: Thank you for raising this. We have rewritten the abstract to focus on the challenges of applying community detection on Bitcoin transaction networks such as the difficulty of interpreting the detected communities due to pseudonymous addresses that do not reveal personal information as opposed to social networks that show users personal details.

Comment 2:  The Introduction is satisfactory. However, the first paragraph requires more clarification.

Reply: Thank you for the positive feedback. We have revised the first paragraph of the introduction accordingly.

Comment 3: “Although the technology can be used in various domains, it is most used in cryptocurrency.”

--> To clarify, the blockchain has been used in many domains for a few years now.

Reply: Thank you for the comment. We have rephrased the statement, and in the preliminaries, we added use cases such as voting systems and land management systems.

Comment 4: “Blockchain transactions are publicly available, but users are linked to pseudonymous addresses which make them often remain unknown.”

--> A transaction does not always mean a cryptocurrency transaction. In a permissioned blockchain, the transactions are not publicly available. Thus, it would be nice to indicate the difference between different types of blockchains.

Reply:  Thank you for the comment. In the introduction, first paragraph, we clarified the difference between permissioned and permissionless blockchains. In the preliminaries, we provided examples of different forms of a transaction.

Comment 5: Section 2 needs improvement with more context.

A background section is needed to define the concepts more clearly.

Reply:  Thank you for pointing out this, we have added an introductory background in section 2 (preliminaries) rather than directly defining the concepts. The new content is in a blue color.

Comment 6:  Section 4 requires more clarification on the experiments and results. Also, the Figures are too small and need improvement.

Reply:  Thank you for pointing out this. The figures have been revised and their size has been increased for improved readability. The flow has been updated with a clarification such as the introductory paragraph that highlights the aim of the experiments. Updated contents are in blue.

Comment 7: Such a broad definition of the Conclusion is not necessary. The Conclusion should be concise and solely focus on discussing the research output and future scope of the work.

Reply: We could not identify the paragraph that is being mentioned in this comment. Regarding future directions, in addition to adapting distance-based clustering methods to improve their performance on disconnected networks to improve their performance on blockchain transaction networks, we highlight that obtaining a larger dataset of illegal addresses could help identify more illegal communities from daily transactions.

Comment 8: Overall, the paper lacks clarity. The authors need to define the concepts and methods more clearly for readers to understand the contribution fully. 

Reply:  We have revised all sections that were mentioned in the comments, we hope that we have addressed them.

Round 2

Reviewer 2 Report

Thanks to the authors for applying the changes. The paper has improved a lot. 

Some minor comments, 

1. The caption of the figures should be concise. The figures should be explained in the text. 

2. The following papers, related to blockchain transactions & consensus, from MDPI can be cited in the Introduction and Preliminaries Section. 

- Assessing Blockchain Consensus and Security Mechanisms against the 51% attack, Sarwar Sayeed & Hector Marco Gisbert

- Proof of Adjourn (PoAj): a Novel approach to mitigate blockchain attacks. Sarwar Sayeed & Hector Marco Gisbert